# Efficacy of probiotics in patients with cognitive impairment: A systematic review and meta-analysis

**Miaomiao Ma**[1,2], **Bo Li**[1,2], **Zhi Qu**[1,2], **Shejuan Liu**[1,2*], **Sisi Li**[1,2]

**1** College of Nursing and Health, Henan University, Kaifeng, Henan, China, **2** Department of Chronic Disease Risk Assessment, Henan University, Kaifeng, Henan, China

* 99373940@qq.com

## Abstract

### Objective

To conduct an in-depth exploration of the specific impacts of probiotics and prebiotic supplements on cognitive impairment, it is imperative to also investigate pertinent factors, including the optimal dosage of probiotics for enhancing cognitive function. This investigation is essential for optimizing probiotic interventions to prevent and treat cognitive decline, aimed at preventing and aiding in the treatment of cognitive decline among patients with cognitive impairment.

### Methods

A comprehensive computerized search was conducted across the Embase, PubMed, Web of Science, Cochrane Library, SinoMed, CNKI, Wanfang and WeiPu Data. Studies targeting randomized controlled trials (RCTs) were included. This search covered a timeframe extending from the inception of each database to September 2024. Following an independent process of literature screening, data extraction, and rigorous quality assessment conducted by two investigators, a meta-analysis was performed using Stata 15.0 software.

### Results

A total of ten studies, involving 778 patients, were included in the analysis. The meta-analysis revealed that probiotics were effective in enhancing cognitive function among patients with cognitive impairment, with a standardized mean difference (SMD) of 0.52 (95% CI: 0.07, 0.98; P < 0.001). Subgroup analysis further demonstrated that the largest effect size was observed for studies utilizing the Mini-Mental State Examination (MMSE) scale as the outcome measure (SMD = 0.88). Additionally, the greatest efficacy was associated with single-strain probiotics (SMD = 0.81), and interventions lasting ≤12 weeks exhibited the most pronounced effect (SMD = 0.61).

**Data availability statement:** All relevant data are within the manuscript and its Supporting Information files.

**Funding:** Funder: Science and Technology Development of Henan Province in the year 2023 Project number: 232102310136 The funders had no role in study design, data collection and analysis, decision to publish, or preparation of the manuscript.

**Competing interests:** The authors have declared that no competing interests exist.

## Conclusion

Probiotics have been shown to enhance cognitive function, with a probiotic intervention program featuring a single probiotic strain and a duration of ≤12 weeks demonstrating particularly robust efficacy in improving cognitive function, as assessed by the MMSE scale.

## Introduction

Cognitive impairment is a condition characterized by diminished brain function, resulting from a variety of etiologies. It is typically manifested by declines or deficits in cognitive abilities, including memory loss, language impairment, visuospatial dysfunction, reduced executive functioning, and impaired comprehension. It primarily encompasses mild cognitive impairment (MCI) and dementia [1]. As the global population ages, the incidence of cognitive impairment among the elderly is progressively rising [2]. According to WHO statistics from 2021, the number of individuals worldwide suffering from cognitive disorders has surpassed 55 million. Some studies predict that this figure will climb to 78 million by 2030 and reach 152.8 million by 2050 [3,4]. In China, cognitive impairment has emerged as a pivotal public health concern [5]. Estimates indicate that the proportion of individuals aged 60 and above with cognitive impairment in China increased from 13.26% in 2010 to 18.70% in 2020 [6]. By 2030, the number of dementia patients in China is projected to exceed 16 million. Dementia ranks as the fourth most significant health threat to the elderly, following cardiovascular, cerebrovascular, and neoplastic diseases [7]. As dementia patients often experience disability and dependence, it imposes substantial economic burdens on families and society [8]. In 2018, it was reported to have caused approximately $1 trillion in global economic losses [9]. If unaddressed, dementia poses a substantial obstacle to social and economic development.

Given the gradual progression of cognitive impairment, identifying and developing interventions targeted at early stages of cognitive decline is of utmost importance [10]. Over the past decade, numerous randomized controlled trials (RCTs) have highlighted the potential beneficial effects of dietary interventions in AD, particularly emphasizing the role of probiotic and prebiotic supplements in slowing the progression of AD [11,12]. Modulating the gut-brain axis has emerged as a novel therapeutic strategy for neurodegenerative disorders, including AD [13]. As our understanding of gut microbiota alterations in AD patients deepens, research is increasingly directed towards more precise "gut microbiota-directed" interventions aimed at managing the progression of AD.

According to the consensus definition established by the International Society for the International Scientific Association for Probiotics and Prebiotics (ISAPP), probiotics are defined as "live microorganisms that, when taken in sufficient amounts, provide health benefits to the host." [14], whereas prebiotics are described as "substrates that are selectively utilized by host microorganisms in the gut to confer a health benefit" [15]. In addition, synbiotics are defined as a mixture of live

microorganisms and substrates that selectively stimulate the growth and/or activity of one or a limited number of microorganisms in the gut. They primarily affect the production of neurotransmitters and promote brain health by regulating the gut microbiota and its metabolites [16]. Given their capacity to modulate the structure and composition of the intestinal microbiota and elicit health benefits, probiotics and prebiotic supplements present novel approaches for the prevention or treatment of certain diseases. Indeed, several representative studies have provided compelling evidence supporting the neuroprotective effects of probiotics and prebiotics in neurological disorders [17–19].

With a surge in research interest, numerous reviews have delved into the therapeutic impacts of probiotic supplementation on neurological disorders, including AD. Notably, these include one systematic review [20] and four meta-analyses [21–24]. While these reviews offer a comprehensive overview of the early literature and provide valuable insights for guiding clinical trials, they are not devoid of limitations. For instance, discrepancies exist among the meta-analysis findings: one study reported no enhancement in cognitive function with probiotics [22], whereas the other three demonstrated improvements in cognitive function post-probiotic treatment [21,23,24]. Unassessed publication bias and high between-group heterogeneity may lead to discrepancies with the results of other studies. Importantly, these meta-analyses solely encompassed studies on the benefits of probiotics and prebiotics published prior to 2021. To update the evidence base, we incorporated data from recently published studies [25–28]. Publication bias was also assessed, and sensitivity analyses were performed to improve the reliability of the results. Furthermore, two of the four meta-analyses [21,23] included only a single study and combined MCI and AD as a singular outcome in their quantitative analyses, potentially leading to biased results. The meta-analysis by Krüger J et al. [22], which encompassed only three studies, did not assess for publication bias, which could be a contributing factor to inter-study heterogeneity. González C et al. [23] did not conduct subgroup analyses despite observing high intergroup heterogeneity in their meta-analysis. Additionally, Lv et al. [29] found that probiotics may be more efficacious in enhancing cognitive function in cognitively impaired individuals compared to healthy individuals, but the study did not explore the potential influence of other pertinent factors, such as dosage.

To address these limitations, we conducted a systematic evaluation and meta-analysis of recently published clinical trials, aiming to provide a deeper understanding of the specific effects of probiotics versus prebiotic supplements on cognitive impairment. We also explored other pertinent factors, such as the optimal dosage of probiotics for improving cognitive function, to better develop a suitable probiotic intervention program for clinical use in the prevention and adjunctive treatment of cognitive impairment in patients.

## Materials and methods

### Study design

This systematic evaluation and meta-analysis adhered to the rigorous guidelines outlined in the Preferred Reporting Items for Systematic Reviews and Meta-Analyses (PRISMA).

### Inclusion and exclusion criteria

The inclusion criteria for this systematic review were established according to the PICOS framework (Population, Intervention, Comparison, Outcomes, Study design), as detailed in Table 1. Specifically, the inclusion criteria encompassed:①Study Design: Randomized controlled trials (RCTs) were considered;②Population: Participants had to exhibit cognitive impairment;③Intervention: The intervention group received probiotics, whereas the control group underwent a placebo intervention;④Outcome Measures: Primary focus was on cognitive function. The exclusion criteria were as follows:①Duplicate Literature: Articles that were identical to others in the review were excluded;②Language Restriction: Only Chinese and English language publications were included;③Data Accessibility and Quality: Studies that were unavailable in full text, had ambiguous, incomplete, or unconvertible data, or those that could not be merged with other studies were excluded;④Quality Assessment: Studies rated as grade C in the quality evaluation were not included.

**Table 1. Inclusion criteria for PICOS studies.**

| Parameter | Inclusion criteria |
| --- | --- |
| Participants | Human studies |
| Interventions | Probiotics (MeSH+free terms) |
| Comparisons | Consistent with an experimental group without any probiotic supplementation (including free-text words) |
| Outcomes | cognitive function |
| Study design | Randomized controlled trial |

Notes: MeSH, Medical Subject Headings.

## Search strategy

A comprehensive computerized literature search was conducted across Web of Science, Embase, Pubmed, the Cochrane Library, CNKI, WeiPu Database, Wanfang Database, and SinoMed. The search strategy employed a combination of subject headings (Mesh terms) and free-text terms, covering a timeframe from inception to September 2024. For instance, in Web of Science, the search query was structured as follows: Cognitive Dysfunction (Topic) OR Cognitive Dysfunctions (Topic) OR Dysfunction, Cognitive (Topic) OR Dysfunctions, Cognitive (Topic) OR Cognitive Impairments (Topic) OR Cognitive Impairment (Topic) OR Impairment, Cognitive (Topic) OR Impairments, Cognitive (Topic) OR Cognitive Disorder (Topic) OR Cognitive Disorders (Topic) OR Disorder, Cognitive (Topic) OR Disorders, Cognitive (Topic) OR Mild Cognitive Impairment (Topic) OR Cognitive Impairment, Mild (Topic) OR Cognitive Impairments, Mild (Topic) OR Impairment, Mild Cognitive (Topic) OR Impairments, Mild Cognitive (Topic) OR Mild Cognitive Impairments (Topic) OR Cognitive Decline (Topic) OR Cognitive Declines (Topic) OR Decline, Cognitive (Topic) OR Declines, Cognitive (Topic) OR Mental Deterioration (Topic) OR Deterioration, Mental (Topic) OR Deteriorations, Mental (Topic) OR Mental Deteriorations (Topic) and Probiotics (Topic) OR Probiotic (Topic) and ((((TS=(randomized controlled trial)) OR TS=(randomized)) OR TS=(placebo)) OR TS=(randomised)) OR TS=(random).

## Literature screening and data extraction

Two independent researchers meticulously screened the literature based on predefined inclusion and exclusion criteria, utilizing EndNote software to eliminate duplicates efficiently. In instances where disagreement arose during the screening process, a third researcher was consulted to engage in a collaborative discussion, ultimately determining the consensus on the final screening outcomes. When necessary, authors were reached out to procure crucial data that was integral to the analysis. The extracted literature encompassed the following data points:①Basic Bibliographic Information: This included details such as the authors' names and the publication dates.②Study Population Characteristics: Information pertaining to the age of participants, duration of their illness, and the sample size was recorded.③Intervention Details: For both the control and intervention groups, specifics on the interventions employed, as well as their timing, were documented.④Outcome Metrics and Measurement Data: The outcome indicators and the corresponding measurement data were meticulously noted.

## Literature quality assessment

The quality of the literature was evaluated by two investigators utilizing the assessment tool outlined in the Cochrane Handbook for Systematic Reviews of Interventions, version 5.1.0 [30], specifically tailored for randomized controlled trials. The assessments focused on several key domains: the adequacy of random sequence generation, allocation concealment, blinding of study participants and personnel, blinding of outcome assessors, completeness of outcome data, selective reporting, and other potential sources of bias. The outcomes of these evaluations were categorized as indicating

a "high risk," "low risk," or "unclear risk" of bias. Based on the cumulative assessment across these domains, studies were graded as follows: Grade A: Studies that fully met all the assessment criteria, demonstrating a low risk of bias across all evaluated domains; Grade B: Studies that met the criteria to a substantial degree but showed some concerns in one or more areas, indicating a mixed level of risk; Grade C: Studies that did not meet the assessment criteria adequately, highlighting significant concerns or a high risk of bias in multiple domains.

### Statistical analysis

A meta-analysis was conducted using Stata version 15.0 software. Heterogeneity across the studies was assessed based on both the P-value and the I² statistic. In cases where P > 0.1 and I² ≤ 50%, indicating the absence of significant heterogeneity, a fixed-effects model was chosen for analysis. Conversely, if P ≤ 0.1 and I² > 50%, suggesting substantial heterogeneity, a random-effects model was employed. Additionally, sensitivity analyses and subgroup analyses were conducted for variables that could potentially contribute to heterogeneity, accompanied by a bias assessment. The outcome measures reported in the included studies were continuous variables, and given the utilization of diverse measurement tools, the standard mean difference (SMD) was adopted as the effect size, with a corresponding 95% confidence interval (CI) calculated. To evaluate the robustness and publication bias of the study, sensitivity analyses and funnel plots were utilized.

## Results

### Search results and Literature characteristics

The initial search yielded a total of 968 articles. Following the removal of duplicates, 700 articles remained for further consideration. Upon meticulous review of the titles, abstracts, and full texts, ten articles [25–28,31–36] were ultimately selected for inclusion in this study. Among these, five were published in English and five in Chinese, collectively encompassing 778 patients; 389 patients comprised the intervention group, and 389 comprised the control group. The screening process and the basic characteristics of the included articles are illustrated in Fig 1 and summarized in Table 2, respectively.

### Evaluation of the quality of literature

Among the ten papers included in the analysis, five were classified as quality A(26, 27, 34–36), with an equal number, five, designated as quality B(25, 28, 31–33). The comprehensive results of the risk of bias assessment conducted on these studies are meticulously outlined in Table 3.

### Meta-analysis results

**Overall inspection.** Ten studies reported on the effectiveness of probiotics in individuals with cognitive impairment, utilizing the standardized mean difference (SMD) as the metric for the combined effect size. Analysis revealed significant heterogeneity among the findings (I² = 88.5%, P < 0.001), prompting the application of the random-effects model for further analysis. As illustrated in Fig 2, the pooled effect size for probiotics in enhancing cognitive function was SMD = 0.52 (P < 0.001), with a 95% confidence interval (CI) ranging from 0.07 to 0.98. This suggests a beneficial role of probiotics in improving cognitive function.

**Subgroup analyses.** Separate subgroup analyses were undertaken, employing the type of cognitive assessment scale (MMSE, ADAS-cog, MoCA, and RBANS), the classification of probiotics (single-species versus multi-species probiotics), and the duration of intervention (≥12 weeks versus <12 weeks) as the respective subgrouping criteria (Table 4).

(1) Subgroup Analysis by Scale Type: Six studies [25,26,28,32–34] utilized the MMSE as the primary outcome measure, while two studies [27,31] employed the Alzheimer's Disease Assessment Scale-Cognitive Subscale (ADAS-cog), and

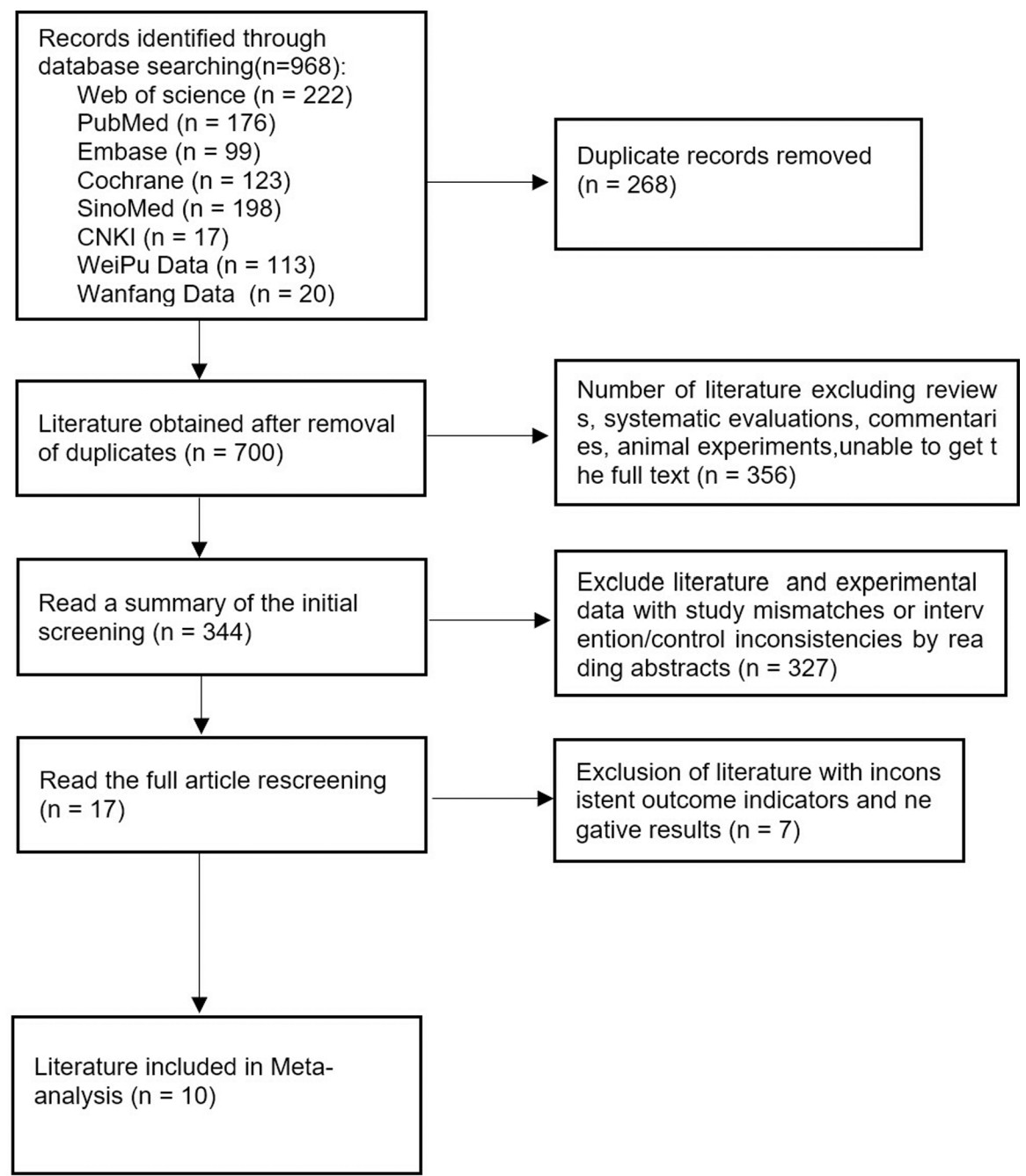

**Fig 1. PRISMA flow chart for the study selection process.**

another two studies [35,36] used the RBANS. The three subgroups exhibited substantial heterogeneity ($I^2 = 87.9\%$, $p < 0.001$), hinting at an influence of the scale type on the association between probiotics and cognitive impairment. Notably, the MMSE scale produced the largest effect size SMD = 0.88 (P = 0.01), 95% CI = (0.67, 1.10) for the outcome of probiotic intervention for cognitive impairment. This was followed by the RBANS scale with an effect size

**Table 2. Basic characteristics of the included literature.**

| Study | Year | Country | Sample size (Intervention group/control group) | Average age (Intervention group/Control group) | Intervention | | Dura-tion | Measurement tools for outcome indicators |
|-------|------|---------|------------------------------------------------|------------------------------------------------|--------------|--|-----------|------------------------------------------|
| | | | | | Intervention group | Control group | | |
| Ma Li | 2021 | China | 30/30 | 71±6/71±6 | Galantamine Tablets + Probiotics | Galanthamine tablets + placebo | 16w | ADAS-cog |
| Wu Baifu | 2020 | China | 54/53 | 60.03±10.29/ 60.42±10.16 | Low frequency repetitive transcranial magnetic stimulation+ probiotics | Low frequency repet-itive transcranial magnetic stimulation | 4w | MMSE |
| Wang Xiaodong | 2015 | China | 14/14 | 67.33±13.08/ 68.32±12.36 | Shimotang Oral Liq-uid + Jin Shuangqi Oral Treatment | Shimotang Oral Liquid | 12m | MMSE |
| He Xianyan | 2022 | China | 40/40 | 70.85±4.47/ 71.02±4.31 | Donepezil + Probiotics | Donepezil | 3m | MMSE+ADAS-Cog |
| Wang Jing | 2022 | China | 29/30 | 74.97±6.78/ 72.21±8.74 | Donepezil Tab-lets + Bifidobacterium Triplex Capsules | Donepezil tablets + placebo | 12w | MMSE |
| Xiao, Jinzhong | 2020 | Japan | 39/39 | 61.3(7.7)/60.9(6.9) | Probiotics | Placebo | 16w | RBANS |
| Daisuke Asaoka | 2022 | Japan | 55/60 | 77.2/78.9 | Probiotics | Placebo | 24w | MMSE+ADAS-Jcog |
| Elmira Akbari | 2016 | Iran | 26/26 | 77.67±2.62/ 82.00±1.69 | Probiotic milk | Milk | 12w | MMSE |
| Y. Kobayashi | 2019 | Japan | 59/58 | 61.5(6.83)/ 61.6 (6.37) | Probiotics | Placebo | 12w | MMSE+ RBANS |
| Yuzhe Fei | 2023 | China | 20/20 | 76.40±9.61/ 75.30±9.75 | Probiotics | Placebo | 12w | MMSE+MoCA |

Notes: w, Week; m, Month; ADAS-cog, Alzheimer disease assessment scale-cog; MMSE, Mini-Mental State Examination; RBANS, Repeatable Battery for the Assessment of Neuropsychological Status; MoCA, Montreal Cognitive Assessment.

SMD = 0.39 (P = 0.002), 95% CI=(0.10, 0.68). the ADAS-cog scale produced the smallest effect size, SMD = -0.31 (P = 0.001), 95% CI=(-0.61, -0.001).

(2) Subgroup Analysis by Probiotic Class: Eight studies [25,27,28,31,33–36] involved complex probiotics, whereas two studies [26,32] focused on single-species. High heterogeneity was observed between these subgroups (I²= 91.2%, P < 0.001), indicating that the class of probiotics may influence the efficacy of interventions for cognitive impairment. Single probiotics produced the largest effect size in improving cognitive function SMD = 0.81 (P = 0.019), 95% CI = (0.37, 1.26). Complex probiotics produced a smaller effect size SMD = 0.17 (P < 0.001), 95% CI= (0.01, 0.33).

(3) Subgroup Analysis by Intervention Duration: The shortest intervention period in the studies was 4 weeks [33], while the longest lasted for 12 months [32]. For the purpose of this subgroup analysis, the extreme values were excluded, and the median duration was adopted as the representative intervention period for inclusion in the analysis. Conse-quently, six studies [25,26,28,33–35] featured an intervention duration of ≤ 12 weeks, and four studies [27,31,32,36] had an intervention duration exceeding 12 weeks. High heterogeneity was noted in the effect sizes between these two subgroups (I²= 88.2%, P < 0.001), suggesting that the length of the intervention period may have a moderate influence on the effectiveness of probiotic interventions in improving cognitive function. Specifically, intervention period ≤12 weeks produced the largest effect size SMD = 0.61 (P < 0.001), 95% CI = (0.42, 0.80) in improving cognitive function. Conversely, intervention period >12 weeks had a smaller effect size SMD = 0.17 (P < 0.001), 95% CI = (-0.07, 0.42).

**Table 3. Quality evaluation of the literature.**

| Study | Random sequence generation | Assignment hiding | Blinding | Completeness of outcome data (follow-up shedding with or without instructions) | Selective reporting of research findings | Additional sources of bias | Quality grade (level) |
|---|---|---|---|---|---|---|---|
| Li Ma [31] | Low risk | Unclear | High risk | Low risk | Low risk | Low risk | B |
| Baifu Wu [33] | Low risk | Unclear | High risk | Low risk | Low risk | Low risk | B |
| Xiaodong Wang [32] | Unclear | Unclear | High risk | Low risk | Low risk | Low risk | B |
| Xianyan He [25] | Low risk | Unclear | High risk | Low risk | Low risk | Low risk | B |
| Jing Wang [26] | Low risk | Low risk | Low risk | Low risk | Low risk | Low risk | A |
| Xiao, Jinzhong [36] | Low risk | Low risk | Low risk | Low risk | Low risk | Low risk | A |
| Daisuke Asaoka [27] | Low risk | Low risk | Low risk | Low risk | Low risk | Low risk | A |
| Elmira Akbari [34] | Low risk | Low risk | Low risk | Low risk | Low risk | Low risk | A |
| Y. Kobayashi [35] | Low risk | Low risk | Low risk | Low risk | Low risk | Low risk | A |
| Yuzhe Fei [28] | Low risk | Low risk | High risk | Low risk | Low risk | Low risk | B |

**Publication bias and sensitivity.** The funnel plot analysis revealed certain asymmetry in the results, potentially suggesting the presence of publication bias within the study. A sensitivity analysis was conducted, involving the sequential exclusion of individual studies to compare changes in the combined outcomes. The analysis indicated that the changes in the results were insignificant, thereby demonstrating the robustness and stability of the study findings (Fig 3 and Fig 4).

## Discussion

Among the ten studies included in this research, five were rated as quality A and five as quality B, all considered to be of high quality. Nine of these studies [25–28,31,33–36] detailed specific methods for generating randomized sequences. Six studies [26–28,34–36] explicitly described the method used for allocation concealment in their text, while five studies [26,27,34–36] specified the blinding procedure for study participants or the individuals administering/measuring the intervention. The consistency of our study with previous studies in terms of the positive effects of probiotics compared to theirs further validates the effectiveness of probiotics in promoting health. However, this study included additional literature and study subjects, and assessed publication bias and performed sensitivity analyses to further enhance the reliability of the results. The included literature comprehensively reported both primary and secondary outcome indicators. The potential sources of heterogeneity in this meta-analysis can be attributed to two factors: firstly, the subjective nature of cognitive functioning scales may contribute to increased heterogeneity in responses related to cognitive functioning effects. Secondly, the study participants hailed from diverse countries and utilized locally adapted scales to assess intervention effects.

In our study, we synthesized data from recent randomized controlled trials examining the use of probiotic and prebiotic supplements in patients with cognitive impairment. The results indicated that, compared to placebo or control treatments, probiotic supplementation exhibited statistically significant positive effects on cognitive function among these patients. Additionally, to delve deeper into the substantial heterogeneity observed among the included studies, we conducted a subgroup analysis. This analysis unveiled correlations between the magnitude of cognitive improvement and factors such as the type of probiotic strain (single versus combination), the duration of the intervention, and the specific cognitive assessment scale utilized. Collectively, these novel findings align with prior research documenting the neuroprotective benefits of probiotics and prebiotics in neurological disorders [17–19].

The findings of this meta-analysis indicate that probiotics can enhance cognitive function in individuals with cognitive impairment. Several studies [21,37,38] have demonstrated that ingesting probiotics or prebiotics can modulate cognitive

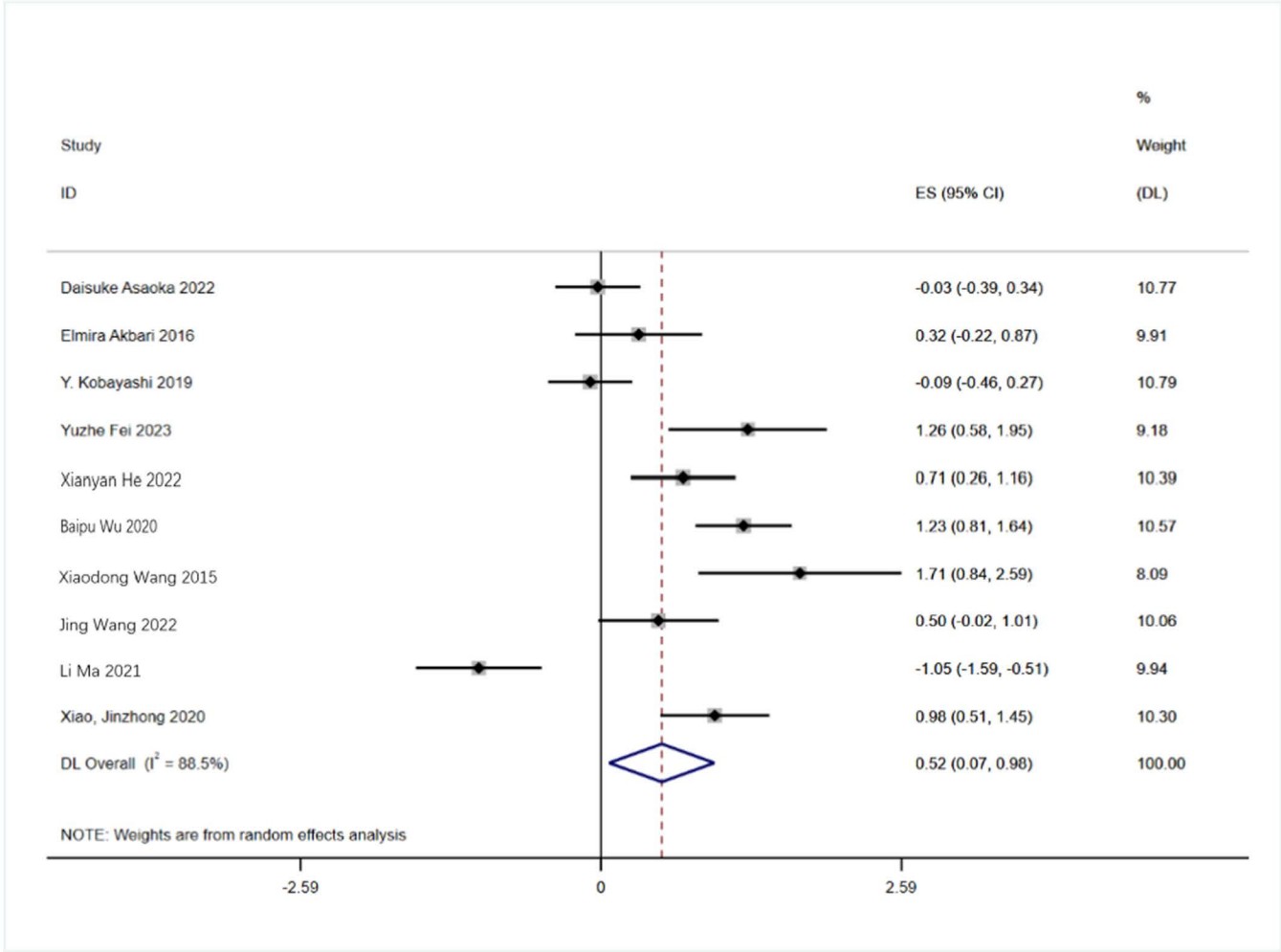

**Fig 2. Forest plot of the overall effect of probiotics on cognition in a random-effects model.**

**Table 4. Subgroup analysis results.**

| Subgroup variable | Categorization | Number of publications (articles) | Effect size and 95% CI | Heterogeneity test | |
|---|---|---|---|---|---|
| | | | | I²(%) | P-value |
| Scale | MMSE | 6 | 0.88 [0.67, 1.10] | 66.6% | P=0.01 |
| | ADAS-cog | 2 | -0.31 [-0.61,-0.001] | 90.6% | P < 0.01 |
| | RBANS | 2 | 0.39 [0.10,0.68] | 89.6% | P < 0.01 |
| Probiotics types | Probiotic complex | 8 | 0.17 [0.01,0.33] | 92.2% | P < 0.01 |
| | Mono-probiotic | 2 | 0.81 [0.37,1.26] | 81.9% | P=0.019 |
| Intervention period | ≤12 weeks | 6 | 0.61 [0.42,0.80] | 79.5% | P < 0.01 |
| | >12 weeks | 4 | 0.17 [-0.07,0.42] | 93.2% | P < 0.01 |

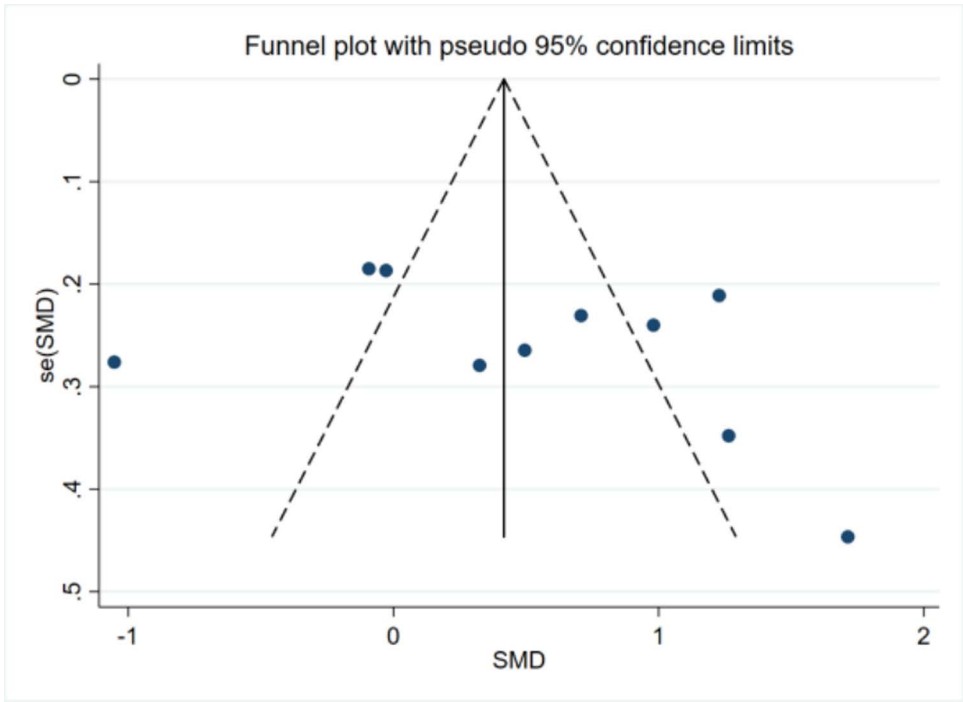

**Fig 3. Funnel plot.**

function. This effect is mediated by the gut microbiota, which reduces inflammatory and oxidative biomarkers. These interactions occur through the microbiota-gut-brain axis. The gut-brain axis is a complex communication system between the gut microbiota and the central nervous system, influencing brain function and behavior through neural, immune, and endocrine pathways. Key mechanisms include neurotransmitter regulation (e.g., serotonin, dopamine, and GABA), which can impact mood and cognitive states; inflammation and immune modulation, where gut dysbiosis can trigger systemic inflammation and neuroinflammation; and direct neural signaling via pathways like the vagus nerve. Probiotics and prebiotics can improve cognitive outcomes by reducing inflammation, regulating neurotransmitters, and enhancing neural signaling. Dietary interventions facilitated by the microbiota-gut-brain axis are being considered as a promising therapeutic approach for cognitive impairment [39,40]. Tao Na and colleagues [41] showed that the integration of probiotics into conventional treatment improved cognitive function, along with lipid and blood glucose profiles, in patients with cognitive impairment.

Probiotics also exhibit the capability to ameliorate symptoms such as anxiety and depression, ultimately enhancing cognitive function. Over the past decade, a continuous influx of novel research has highlighted the favorable effects of dietary probiotic interventions on anxiety, depression, and cognitive function in various disease states [42]. For instance, a randomized, double-blind, placebo-controlled clinical trial [43] revealed that a daily probiotic intake for 12 consecutive days significantly elevated MDS-UPDRS (Unified Parkinson's Disease Rating Scale) scores among patients with Parkinson's disease. Furthermore, a recent study [18] noted a slight but statistically significant improvement in depression and anxiety symptoms with daily probiotic consumption, whereas no such effect was observed with prebiotic intake. Other studies have provided evidence that similarly supports the positive impact of probiotic therapy on disorders linked to the gut-brain axis. A randomized controlled trial [44] demonstrated that healthy volunteers treated with a probiotic mixture had a significantly reduced response to sadness. This effect was observed compared to a placebo group over a 4-week intervention period. Another study conducted on healthy individuals [45] found that consuming yogurt containing probiotics for

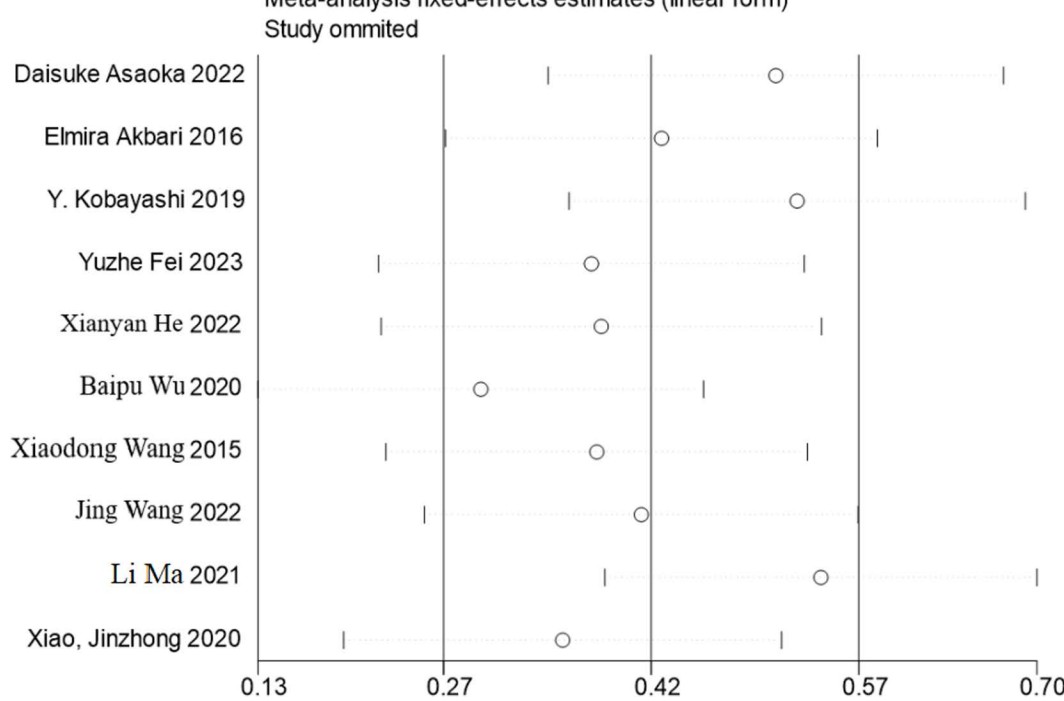

**Fig 4. Sensitivity analysis.**

3 consecutive weeks significantly improved mood states. Additionally, an earlier study on irritable bowel syndrome (IBS) [46] revealed beneficial systemic and immunomodulatory effects of probiotics, with a notable normalization of the interleukin-10 to interleukin-12 ratio observed exclusively in IBS patients treated with Bifidobacterium infantis 35624. These findings suggest that the immunologic benefits of probiotics are not only extensive but also exhibit more pronounced strain-specificity than previously recognized [47].

Furthermore, given the variability in the severity of the participants enrolled in this study, the choice of an appropriate scale for assessing cognitive impairment was of paramount importance. The results of our subgroup analysis, which included only the MMSE, ADAS-cog, and RBANS scales, indicated that the MMSE scale yielded the largest effect size in evaluating cognitive improvement, implying its superiority over the other scales in this regard. In a study by D.A. Loewenstein et al. [48] involving older adults with an average age of 75 years or older, it was found that when using a MMSE cut-off value of ≤26 points, the sensitivity for distinguishing MCI from normal aging was 70.8%, with a specificity of 84.6%. The authors concluded that while the MMSE was effective in differentiating cognitive impairment from normal aging, it was less discriminatory between normal aging and MCI. Consequently, future clinical studies should consider the influence of multiple factors when selecting assessment tools, aiming to more precisely evaluate the impact of probiotics on cognitive functioning, thereby accurately assessing the intervention effect of probiotics.

The impact of probiotic species and the duration of the intervention on cognitive enhancement varies. The findings of this study suggest that a single probiotic species exerts the most significant effect on improving cognition in individuals with cognitive dysfunction. This aligns with the observations made by Tingting Lv et al. [29], who demonstrated that supplementation with a single probiotic species exhibited a more pronounced therapeutic effect. This may be attributed to the antagonistic interactions and competition for gut nutrients among composite probiotic species, which could potentially diminish their efficacy [49]. Furthermore, our study results indicate that intervention periods of ≤12 weeks yield the

greatest improvement in cognitive function. The difference in cognitive enhancement observed with intervention cycles exceeding 12 weeks was not statistically significant, contrasting with the findings of Nanyang Liu et al. [50], who reported that supplementation durations longer than 12 weeks were effective in enhancing cognitive performance. The discrepancy in results may stem from the inclusion of both Chinese and English literature in our analysis. Additional studies are required in the future to further validate these findings.

Research findings indicate that various probiotics can ameliorate cognitive dysfunction, and a consensus statement published by the ISAPP [14] suggests that adequate probiotic supplementation may offer certain health benefits. However, ISAPP does not delineate specific dosages and frequencies for probiotic supplementation. Over the past decade, entities such as ISAPP [14] and the World Gastroenterology Organisation [51] have endeavors to establish recommended probiotic doses. To ensure the safety and effectiveness of probiotic use, the ISAPP advises that the daily count of live probiotic cells in foods and dietary supplements should be at least $1 \times 10^9$ colony-forming units (CFU). In the literature reviewed for this study, intervention doses could not be standardized in terms of units due to the inclusion of diverse probiotic forms such as capsules, liquids, and others, and some studies [28,35] failed to specify the intervention frequency. Consequently, the available data were insufficient for conducting a subgroup analysis. Future research should endeavor to screen probiotic supplements for optimal dosage and ingestion frequency. Additionally, there is a pressing need for more efficacious probiotic screening methodologies to develop targeted probiotic strategies for cognitive impairment.

The current study is subject to several limitations. Firstly, the literature reviewed in this paper did not exclude the potential interference of other medications or forms of exercise, such as galantamine tablets, donepezil, moderate resistance training, and other factors. These medications and forms of exercise may exert a direct influence on gut microbiota composition and metabolic profiles, which could subsequently impact the gut-brain axis and related disorders. Therefore, it is imperative that future studies account for these variables. Secondly, heterogeneity may arise due to variations in probiotic supplement manufacturers. While probiotics as dietary supplements have been widely deemed safe for use based on prior research, the elderly population is particularly susceptible to serious adverse effects, including gastrointestinal discomfort, systemic infections, and skin issues, owing to their decreased immune function. Furthermore, quality assessment of included studies is based on predetermined criteria, but some subjective judgment may be involved in the process. Synthesizing results is challenging because of the heterogeneity of studies in terms of interventions and outcomes. Finally, this study inevitably faces the challenges of limited sample size and short duration, which undoubtedly limit its applicability. In particular, small sample sizes may weaken the robustness of statistical results and generalizability of conclusions, while short study durations may not provide a comprehensive picture of the long-term trends in the evolution of the study variables or their potential far-reaching implications. In light of this, future studies must improve the reporting of potential adverse effects associated with probiotic supplements [52]. and robiotics can be compared with other interventions to enhance the effectiveness of probiotics as a treatment. By thoroughly considering these factors and addressing the limitations when designing probiotic intervention programs, we can gain a more comprehensive understanding of the potential benefits of probiotics on cognitive impairment. This will facilitate the systematic advancement of research aimed at improving cognitive function with probiotics.

## Conclusions

In In summary, the meta-analysis conducted in this study revealed that probiotics can enhance cognitive abilities in patients with cognitive impairment. Furthermore, this study furnishes a robust scientific rationale for the potential application of probiotics in areas encompassing neurodegenerative diseases and mental health concerns. Future research endeavors should concentrate on refining the dosage and administration protocols of probiotics to ensure the maximization of their salutary effects on brain health. Concurrently, we advocate for interdisciplinary synergy, amalgamating the expertise of neuroscience, microbiology, and clinical medicine to propel advancements in probiotics research pertaining to brain health.

## Supporting information

**S1 Table. PRISMA checklist.**
(DOCX)

**S2 Table. List of raw analysis data.**
(XLSX)

**S3 Table. Study extraction overview table.**
(DOCX)

**S4 Table. Comprehensive study table with inclusion and exclusion details.**
(XLSX)

**S1 File. Quality evaluation of the literature.**
(DOCX)

**S2 File. Search strategy.**
(DOCX)

**S3 File. Retrieval process and data extraction.**
(DOCX)

**S4 File. Data analysis and results.**
(DOCX)

## Acknowledgments

The authors would like to thank all the participants of the study.

## Author contributions

**Supervision:** Bo Li.

**Validation:** Bo Li.

**Writing – original draft:** Miaomiao Ma, Sisi Li.

**Writing – review & editing:** Miaomiao Ma, Bo Li, Zhi Qu, Shejuan Liu, Sisi Li.

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
