## [Decision Letter · Decision Letter 0]

24 Feb 2025

PONE-D-24-49949Efficacy of probiotics in patients with cognitive impairment: A systematic review and meta-analysisPLOS ONE

Dear Dr. Liu,

Thank you for submitting your manuscript to PLOS ONE. After careful consideration, we feel that it has merit but does not fully meet PLOS ONE’s publication criteria as it currently stands. Therefore, we invite you to submit a revised version of the manuscript that addresses the points raised during the review process.

We look forward to receiving your revised manuscript.

Kind regards,

Zohreh Sajadi Hezaveh

Academic Editor

PLOS ONE

Journal Requirements:

2.  As required by our policy on Data Availability, please ensure your manuscript or supplementary information includes the following:

“This study was supported by the Science and Technology Development of Henan Province in the year 2023 (232102310136). The funders were not associated with the conceptualization, design, implementation, or approval of this study.”

6. We note that your Data Availability Statement is currently as follows: All relevant data are within the manuscript and its Supporting Information files.

7.  Please include a separate caption for each figure in your manuscript.

Reviewers' comments:

Reviewer's Responses to Questions

**Comments to the Author**

1. Is the manuscript technically sound, and do the data support the conclusions?

Reviewer #1: Yes

Reviewer #2: Yes

2. Has the statistical analysis been performed appropriately and rigorously? 

Reviewer #1: Yes

Reviewer #2: Yes

3. Have the authors made all data underlying the findings in their manuscript fully available?

Reviewer #1: Yes

Reviewer #2: Yes

4. Is the manuscript presented in an intelligible fashion and written in standard English?

Reviewer #1: Yes

Reviewer #2: Yes

5. Review Comments to the Author

Reviewer #1: In this systematic review and meta-analysis, the impact of probiotics on patients with cognitive impairment was evaluated. Conclusion: Probiotics have been proven to enhance cognitive function, and a probiotic intervention program containing a single probiotic strain with a duration of<12 weeks has shown particularly strong effects in evaluating cognitive function through Mini Mental State Examination (MMSE). Some possible mechanisms have also been proposed in this regard. After careful evaluation, I found that most of the criteria for the review and meta-analysis of this manuscript have been taken into account. However, recheck the manuscript using the PRISMA checklist and revise or complete it to ensure that all standards have been met.

This paper is very interesting, but the following modifications still need to be made.

1.We noticed that the articles included in this review were from the establishment of the database until September 2024. But we noticed that the articles included by the researchers were mostly from after 2015. Based on your search strategy and search time, have you conducted sufficient searches on various databases, and were there any relevant articles before 2015? For what reason was it not included?

2.In the subgroup analysis, separate analyses were conducted on the scales used, intervention time, and types of probiotics. We noticed that the age range of the researchers included in this study was between 50-85 years old. As aging and aging are themselves high-risk factors for cognitive decline, no age-related subgroup analysis was conducted to determine the reasons for this.

3.The author order in Tables 3 and 4 should be based on alphabetical order or the year of publication of the article.

4.In the discussion, we found that sometimes cognitive impairment is used, but later it is also described in patients with Alzheimer's disease. Please clarify and distinguish the concepts of the two, pay attention to the theme of this article, and maintain consistency before and after.

5. In the discussion, if you can compare your findings with other intervention measures and ultimately determine the efficiency of your findings, it will be more convincing

6. This study found that the underlying theory needs a clearer explanation.

Some sentences are too long to understand and keep up with.

Reviewer #2: General Comment: This manuscript addresses an important topic—the effect of probiotics on cognitive impairment—through a systematic review and meta-analysis. The paper is generally well-structured, but there are several areas that need clarification and improvement. Below are my detailed comments:

Abstract:

1. Line 8-10: The sentence "Such an investigation is crucial for developing a tailored probiotic intervention program..." could be made more concise. Consider revising to: "This investigation is essential for optimizing probiotic interventions to prevent and treat cognitive decline."

2. Study Selection Criteria: It would be helpful to briefly mention the inclusion criteria for the meta-analysis in the abstract (e.g., "Only randomized controlled trials (RCTs) were included.").

3. The full term "Mini-Mental State Examination (MMSE)" is spelled out correctly upon first mention (e.g., in line 29). Please use abbreviations on line 36.

Introduction:

4. Line 24-26: The paper mentions "several studies" providing evidence for the neuroprotective effects of probiotics but only lists a limited number of citations (17-19). Consider adding more references or specifying that these are representative studies.

5. Line 30-32: The introduction of synbiotics is relevant but should include a brief explanation of their role in cognitive function to make the connection clearer for readers unfamiliar with the topic.

6. Contradictory Findings: The introduction briefly touches on inconsistencies in previous meta-analyses but does not explore this in enough detail. It would be helpful to discuss why there are discrepancies among prior findings and how your study addresses these issues.

Methods:

7. Inclusion Criteria: The PICOS framework is described well. However, the exclusion criteria (particularly point ③ regarding data accessibility) could benefit from more detail. For instance, how was data ambiguity determined? Were attempts made to contact study authors for missing data?

8. Search Strategy: The search is comprehensive, but the search terms and databases are listed in an unclear manner. Please provide a sample search string (e.g., for Embase, PubMed, Web of Science,) as supplementary material or in an appendix for transparency and reproducibility.

9. Quality Assessment: The use of Cochrane's risk of bias tool is appropriate, but you mention "subjective judgment" in quality assessment. Could you specify if this was mitigated by having two reviewers assess each study independently? Including details about inter-rater agreement would strengthen the validity of the assessment.

Results:

10. Tables 2 and 4: The table presents a good summary, but it could be enhanced by adding a footnote explaining the abbreviations in the intervention column (e.g., MMSE, ADAS-cog). This will help readers unfamiliar with these terms.

Discussion:

11. Line 140-145: The discussion effectively highlights the key findings. However, the limitations of the included studies (such as small sample sizes or short durations) are not sufficiently addressed. A more thorough discussion of these limitations would strengthen the paper.

12. Interpretation of Results: The positive effect of probiotics is well-established, but the discussion could be improved by directly comparing the results with those from prior meta-analyses. Specifically, how do your results differ from the studies by Den et al. (2020) or Krüger et al. (2021)?

13. Line 167-172: The mention of gut-brain axis mechanisms is important but lacks depth. More specific details on the biological mechanisms (e.g., reduction of inflammation, modulation of neurotransmitters) could enhance the discussion.

Conclusion:

14. Line 190-195: The conclusion could be more concise. Consider reducing the repetition of points already made in the discussion. Also, a clearer statement regarding clinical implications and future research directions (e.g., focusing on probiotic dosage) would be useful.

6. PLOS authors have the option to publish the peer review history of their article (what does this mean? ). If published, this will include your full peer review and any attached files.

**Do you want your identity to be public for this peer review?** For information about this choice, including consent withdrawal, please see our Privacy Policy .

Reviewer #1: No

Reviewer #2: No

---

## [Author Response · Author response to Decision Letter 1]

3 Mar 2025

Revision Report

First of all, I would like to express our sincere gratitude to the reviewers for their comments. These comments are all valuable and helpful for revising and improving our manuscript, as well as the important guiding significance to our researches. We have studied comments carefully and have made correction which we hope meet with approval. Revised portions are marked in red in the revised version. The responses to the reviewer’s comments are listed below.

Response to editors and reviewers

(original comments by editors and reviewers are in blue color )

Editors:

1.Comment: Please ensure that your manuscript meets PLOS ONE's style requirements, including those for file naming.

1.Reply: Thank you for your comments. We have made sure that the manuscript meets PLOS ONE's style requirements, including file naming.

2.Comment: As required by our policy on Data Availability, please ensure your manuscript or supplementary information includes the following:A numbered table of all studies identified in the literature search, including those that were excluded from the analyses For every excluded study, the table should list the reason(s) for exclusion If any of the included studies are unpublished, include a link (URL) to the primary source or detailed information about how the content can be accessed�A table of all data extracted from the primary research sources for the systematic review and/or meta-analysis. The table must include the following information for each study:Name of data extractors and date of data extraction Confirmation that the study was eligible to be included in the review.�All data extracted from each study for the reported systematic review and/or meta-analysis that would be needed to replicate your analyses.If data or supporting information were obtained from another source (e.g. correspondence with the author of the original research article), please provide the source of data and dates on which the data/information were obtained by your research group If applicable for your analysis, a table showing the completed risk of bias and quality/certainty assessments for each study or outcome.  Please ensure this is provided for each domain or parameter assessed. For example, if you used the Cochrane risk-of-bias tool for randomized trials, provide answers to each of the signalling questions for each study. If you used GRADE to assess certainty of evidence, provide judgements about each of the quality of evidence factor. This should be provided for each outcome An explanation of how missing data were handled. 

2.Reply�A numbered table of all studies identified in the literature search, including those excluded from the analyses, is provided in the supplementary information (Comprehensive study table with inclusion and exclusion details). For each excluded study, the table lists the specific reason(s) for exclusion. Additionally, some tables of all data extracted from the primary research sources for the systematic review and/or meta-analysis is included in the supplementary information(S2 Table; S3 Table; S1 File; S3 File). No missing data were identified in this study, and all included studies were published.

3.Comment: We suggest you thoroughly copyedit your manuscript for language usage, spelling, and grammar. If you do not know anyone who can help you do this, you may wish to consider employing a professional scientific editing service.

3.Reply Thank you for your suggestion. We appreciate your attention to the language quality of our manuscript. Before submission, the manuscript was thoroughly reviewed and corrected by faculty members specializing in English from our college. Upon receiving feedback, we meticulously checked the language, spelling, and grammar to ensure that each word was spelled correctly and that terminology was consistent and accurate throughout the document. We have also carefully reviewed the manuscript to ensure that the format is consistent, and the language is logical and fluent. We are confident that this rigorous review process has greatly enhanced the professionalism and quality of our manuscript.

4.Comment: We note that the grant information you provided in the ‘Funding Information’ and ‘Financial Disclosure’ sections do not match.When you resubmit, please ensure that you provide the correct grant numbers for the awards you received for your study in the ‘Funding Information’ section.

4.Reply Thank you for bringing this discrepancy to our attention. We apologize for the inconsistency between the grant information provided in the ‘Funding Information’ and ‘Financial Disclosure’ sections. We have carefully reviewed the grant details and will ensure that the correct grant numbers for the awards received for our study are accurately listed in the ‘Funding Information’ section upon resubmission.

5.Comment: Thank you for stating the following financial disclosure:“This study was supported by the Science and Technology Development of Henan Province in the year 2023 (232102310136). The funders were not associated with the conceptualization, design, implementation, or approval of this study.”Please state what role the funders took in the study.  If the funders had no role, please state: ""The funders had no role in study design, data collection and analysis, decision to publish, or preparation of the manuscript.""If this statement is not correct you must amend it as needed.Please include this amended Role of Funder statement in your cover letter; we will change the online submission form on your behalf.

5.Reply Thank you for your suggestion. We have made the requested modifications and revised the funder role statement as follows: "The funders had no role in study design, data collection and analysis, decision to publish, or preparation of the manuscript." This revised statement has been included in the cover letter to ensure clarity and consistency. 

6.Comment: We note that your Data Availability Statement is currently as follows: All relevant data are within the manuscript and its Supporting Information files.Please confirm at this time whether or not your submission contains all raw data required to replicate the results of your study. Authors must share the “minimal data set” for their submission. PLOS defines the minimal data set to consist of the data required to replicate all study findings reported in the article, as well as related metadata and methods(https://journals.plos.org/plosone/s/data-availability#loc-minimal-data-set-definition).

6.Reply The authors have confirmed that the submitted file contains all the raw data needed to replicate the results of the study. All relevant data are within the paper and its Supporting Information files.

7.Comment: Please include a separate caption for each figure in your manuscript.

7.Reply Thank you for your careful review of the manuscript. During the manuscript preparation stage, we assigned a descriptive title to each image, with the aim of clearly and accurately conveying the content and core message of the figure. Upon receiving your feedback, we have meticulously reviewed and correctly cited each figure title in the manuscript to ensure consistency with the overall style and formatting requirements. We have also verified that all terminology is accurate and consistent throughout the document to enhance the clarity and coherence of the manuscript.

8.Comment: Please include captions for your Supporting Information files at the end of your manuscript, and update any in-text citations to match accordingly. 

8.Reply In response to your request, we have appended the titles of all supporting information documents at the conclusion of the manuscript. These documents encompass, but are not confined to: S1 Table. PRISMA checklist; S2 Table. List of raw analysis data; S3 Table. Study extraction overview table; S1 File. Quality evaluation of the literature; S2 File.Search strategy; S3 File.Retrieval process and data extraction; S4 File. Data analysis and results. etc. To facilitate readers in locating and citing pertinent supporting information documents, we have thoroughly updated all references to these documents within the main text of the manuscript.

Reviewer #1

1.Comment: We noticed that the articles included in this review were from the establishment of the database until September 2024. But we noticed that the articles included by the researchers were mostly from after 2015. Based on your search strategy and search time, have you conducted sufficient searches on various databases, and were there any relevant articles before 2015? For what reason was it not included?

1.Reply In response to your queries, we conducted a comprehensive review. A total of 60 articles published before 2015 were excluded because they did not meet the inclusion criteria. These criteria included: (1) articles reporting animal experiments; (2) studies whose content did not match our research focus; and (3) literature classified as reviews or systematic reviews. The specific reasons for excluding each article are detailed in the supporting information (Comprehensive study table with inclusion and exclusion details). For example, the 2007 article by D. Benton titled “Impact of consuming a milk drink containing a probiotic on mood and cognition” was excluded because it examined the effects of drinking milk containing probiotics on mood and was conducted in healthy adults. Additionally, research on probiotics and cognitive impairment has increased significantly since 2016. Specifically, there were 12 publications in 2014, 10 in 2015, and 27 in 2016. By 2020, the number had reached 75. We hope this response clarifies the rationale behind our article selection process.

2.Comment: In the subgroup analysis, separate analyses were conducted on the scales used, intervention time, and types of probiotics. We noticed that the age range of the researchers included in this study was between 50-85 years old. As aging and aging are themselves high-risk factors for cognitive decline, no age-related subgroup analysis was conducted to determine the reasons for this.

2.Reply Thank you for your insightful comment. We have conducted an in-depth discussion regarding the questions you raised concerning age-related subgroup analysis. As you aptly noted, the participants in this study spanned an age range from 50 to 85 years, with aging itself being a recognized risk factor for cognitive decline. When performing subgroup analyses, our primary focus was on factors such as the scales employed, the duration of intervention, and the type of probiotics used, and we were unable to perform a more nuanced stratification analysis for age.

To further elaborate and clarify:

Firstly, during the initial design phase of this study, our primary emphasis was on assessing the impact of probiotics on cognitive ability. Based on this focus, we developed a corresponding subgroup analysis plan.

Secondly, the limited number of studies included in our analysis, coupled with variations in age distribution among these studies, may have contributed to non-conclusive findings from the subgroup analyses.

Lastly, the age-related subgroup analysis you proposed is indeed intriguing. In future research endeavors, we will give greater attention to the variable of age and endeavor to collect data from larger sample sizes to facilitate more detailed and in-depth subgroup analyses.

Your feedback is highly valued, and we will take these considerations into account in our future work.

3.Comment: The author order in Tables 3 and 4 should be based on alphabetical order or the year of publication of the article.

3.Reply We agree with your suggestion and have accordingly arranged the authors listed in Tables 2 and 3 in alphabetical order.(Table 4 represents a subgroup analysis and does not require alphabetical sorting.)

4.Comment: In the discussion, we found that sometimes cognitive impairment is used, but later it is also described in patients with Alzheimer's disease. Please clarify and distinguish the concepts of the two, pay attention to the theme of this article, and maintain consistency before and after.

4.Reply: In response to your review comment, we acknowledge the need for clarification and distinction between the concepts of "cognitive impairment" and "Alzheimer's disease" within our discussion. It is important to maintain consistency with the theme of our paper.To clarify, On the one hand,cognitive impairment refers to a broad range of symptoms that affect mental functioning, including memory, language, and problem-solving abilities. These impairments can be mild, moderate, or severe and may be caused by various factors, including aging, brain injuries, or underlying medical conditions.On the other hand, Alzheimer's disease is a specific type of dementia that is characterized by progressive decline in memory, thinking, and behavioral abilities. It is a chronic neurodegenerative disorder that affects multiple brain regions and eventually leads to severe cognitive and functional impairment.

In our discussion, we may have used the term "cognitive impairment" in a general sense to describe the overall effects of the intervention on cognitive functioning. However, when specifically referring to studies that focused on Alzheimer's disease patients, we have clearly identified them as such.

To ensure consistency with the theme of our paper,We have revised the terminology in the discussion section to maintain consistency of terminology and themes before and after.We will ensure that our terminology is accurate and that our discussion remains focused on the relevant aspects of our research.

Thank you for bringing this to our attention, and we appreciate your constructive feedback.

5.Comment: In the discussion, if you can compare your findings with other intervention measures and ultimately determine the efficiency of your findings, it will be more convincing

5.Reply We fully understand the significance of such comparisons in enhancing the persuasiveness of our discussion.However, it is important to note that various intervention measures, including probiotics, often have their specific application scopes and are utilized based on the unique characteristics and needs of individual patients. In the case of probiotics, while they have shown promising results as an adjunctive therapy in numerous studies, they are indeed rarely used as a standalone treatment.

Given this context, we acknowledge that a direct comparison of probiotics with other primary intervention measures may not always be straightforward or entirely applicable. Instead, we have focused on discussing the potential benefits and limitations of probiotics as an adjunctive therapy, highlighting their role in supporting the overall treatment regimen.

We will carefully consider these points and add to the discussion accordingly to ensure that our findings are presented in a comprehensive and persuasive manner.

6.Comment: This study found that the underlying theory needs a clearer explanation.Some sentences are too long to understand and keep up with.

6.Reply�I apologize for any difficulty caused by the lengthy sentences in the manuscript. In response to your feedback, Firstly�we have explained the concepts of grounded theory in the manuscript in a more detailed and clearer way, for example, the sentence "Cognitive impairment encompasses the decline in one or more cognitive domains, including memory, language, visuospatial abilities, executive functions, computation, and comprehension and judgment" modified to "Cognitive impairment is a condition characterized by diminished brain function, resulting from a variety of etiologies". Secondly We have reviewed the entire manuscript and revised the long sentences to improve clarity and readability. For instance�the sentence "Several studies(21, 37, 38) have demonstrated that the ingestion of probiotics or prebiotics enables the gut microbiota to regulate cognitive function by reducing inflammatory and oxidative biomarkers via interactions with the brain through the microbiota-gut-brain axis." modified to "Several studies(21, 37, 38) have demonstrat

---

## [Editor Report · Decision Letter 1]

9 Mar 2025

Efficacy of probiotics in patients with cognitive impairment: A systematic review and meta-analysis

PONE-D-24-49949R1

Dear Dr. Liu,

We’re pleased to inform you that your manuscript has been judged scientifically suitable for publication and will be formally accepted for publication once it meets all outstanding technical requirements.

Kind regards,

Zohreh Sajadi Hezaveh

Academic Editor

PLOS ONE
---

## [Editor Report · Acceptance letter]

PONE-D-24-49949R1

PLOS ONE

Dear Dr. Liu,

I'm pleased to inform you that your manuscript has been deemed suitable for publication in PLOS ONE. Congratulations! Your manuscript is now being handed over to our production team.

Kind regards,

on behalf of

Dr. Zohreh Sajadi Hezaveh

Academic Editor

PLOS ONE